# Developmental Roles of FUSE Binding Protein 1 (*Fubp1*) in Tooth Morphogenesis

**DOI:** 10.3390/ijms21218079

**Published:** 2020-10-29

**Authors:** Yam Prasad Aryal, Sanjiv Neupane, Tae-Young Kim, Eui-Seon Lee, Nitin Kumar Pokhrel, Chang-Yeol Yeon, Ji-Youn Kim, Chang-Hyeon An, Seo-Young An, Eui-Kyun Park, Jung-Hong Ha, Jae-Kwang Jung, Hitoshi Yamamoto, Sung-Won Cho, Sanggyu Lee, Do-Yeon Kim, Tae-Yub Kwon, Youngkyun Lee, Wern-Joo Sohn, Jae-Young Kim

**Affiliations:** 1Department of Biochemistry, School of Dentistry, IHBR, Kyungpook National University, 2177, Dalgubeol-daero, Jung-gu, Daegu 41940, Korea; yparyal86@gmail.com (Y.P.A.); tae09290@daum.net (T.-Y.K.); euiseon3488@gmail.com (E.-S.L.); nitinpokhrel.np@gmail.com (N.K.P.); yhs2669@naver.com (C.-Y.Y.); ylee@knu.ac.kr (Y.L.); 2Department of Biochemistry and Cell Biology, Stony Brook University, Stony Brook, NY 11784, USA; sanjiv.knu@gmail.com; 3Department of Dental Hygiene, Gachon University, Incheon 21936, Korea; hoho6434@gachon.ac.kr; 4Department of Oral and Maxillofacial Radiology, School of Dentistry, IHBR, Kyungpook National University, 2177, Dalgubeol-daero, Jung-gu, Daegu 41940, Korea; chan@knu.ac.kr (C.-H.A.); syan@knu.ac.kr (S.-Y.A.); 5Department of Oral Pathology and Regenerative Medicine, School of Dentistry, IHBR, Kyungpook National University, 2177, Dalgubeol-daero, Jung-gu, Daegu 41940, Korea; epark@knu.ac.kr; 6Department of Conservative Dentistry, School of Dentistry, IHBR, Kyungpook National University, 2177, Dalgubeol-daero, Jung-gu, Daegu 41940, Korea; endoking@knu.ac.kr; 7Department of Oral Medicine, School of Dentistry, IHBR, Kyungpook National University, 2177, Dalgubeol-daero, Jung-gu, Daegu 41940, Korea; widenmy@knu.ac.kr; 8Department of Histology and Developmental Biology, Tokyo Dental College, Tokyo 101-0061, Japan; hyamamoto@tdc.ac.jp; 9Division in Anatomy and Developmental Biology, Department of Oral Biology, Yonsei University College of Dentistry, Seoul 26493, Korea; chosome1@yuhs.ac; 10School of Life Science BK21 plus KNU Creative BioResearch Group, Kyungpook National University, Daegu 41566, Korea; slee@knu.ac.kr; 11Department of Pharmacology, School of Dentistry, Kyungpook National University, 2177, Dalgubeol-daero, Jung-gu, Daegu 41940, Korea; dykim82@knu.ac.kr; 12Department of Dental Biomaterials, School of Dentistry, Kyungpook National University, 2177, Dalgubeol-daero, Jung-gu, Daegu 41940, Korea; tykwon@knu.ac.kr; 13Pre-Major of Cosmetics and Pharmaceutics, Daegu Haany University, Gyeongsan 38610, Korea; wjsohn@dhu.ac.kr

**Keywords:** *Fubp1*, dentinogenesis, amelogenesis, transcriptional regulator, tooth development

## Abstract

FUSE binding protein 1 (*Fubp1*), a regulator of the c-Myc transcription factor and a DNA/RNA-binding protein, plays important roles in the regulation of gene transcription and cellular physiology. In this study, to reveal the precise developmental function of *Fubp1*, we examined the detailed expression pattern and developmental function of *Fubp1* during tooth morphogenesis by RT-qPCR, in situ hybridization, and knock-down study using in vitro organ cultivation methods. In embryogenesis, *Fubp1* is obviously expressed in the enamel organ and condensed mesenchyme, known to be important for proper tooth formation. Knocking down *Fubp1* at E14 for two days, showed the altered expression patterns of tooth development related signalling molecules, including *Bmps* and *Fgf4*. In addition, transient knock-down of *Fubp1* at E14 revealed changes in the localization patterns of c-Myc and cell proliferation in epithelium and mesenchyme, related with altered tooth morphogenesis. These results also showed the decreased amelogenin and dentin sialophosphoprotein expressions and disrupted enamel rod and interrod formation in one- and three-week renal transplanted teeth respectively. Thus, our results suggested that *Fubp1* plays a modulating role during dentinogenesis and amelogenesis by regulating the expression pattern of signalling molecules to achieve the proper structural formation of hard tissue matrices and crown morphogenesis in mice molar development.

## 1. Introduction

Organogenesis is a complex phenomenon by which cells of different embryonic origin develop and associate to form organs [1]. Various organs, such as limb, mammary gland, kidney, feather, tooth, hair, lung, and pancreas, share common features, for instance, epithelial–mesenchymal interactions, during the early stages of organogenesis [2,3,4]. The interacting signalling from tissues modulates cell proliferation, differentiation, apoptosis and migration- during morphogenesis and regeneration [5]. Particularly, tooth develops through complex signalling regulations from reciprocal epithelial–mesenchymal interactions with unique and dynamic structural alterations. In mice, the thickening of the epithelium and its gradual invagination towards mesenchyme indicate the morphological commencement of odontogenesis [6]. Following epithelial invaginations, through the combined interactions of signalling molecules and transcription factors, tooth development proceeds with bud, cap, and bell stages at mouse embryonic stages E13, 14, and 16, respectively, which shows typical morphologic features of epithelial appendages with epithelial invaginations at E13, signalling centre formation with specific epithelial structural formation of enamel knot at E14, and functional differentiation of odontoblasts and ameloblasts at E16 [6,7]. There are about 500 genes involved in tooth development [8] and the spatio-temporal expression patterns that characterize the developmental function of these genes are important for proper tooth morphogenesis. The signalling pathways for proper morphogenesis during the developmental stages are strictly regulated at the transcription step which controls the paracrine factors, including *Shh*, *Bmp*, *Fgf*, and *Wnt* signalling [9]. For proper modulation of these paracrine factors, it is necessary to understand the developmental roles of transcriptional regulators, which modulate the expression of target genes and contribute to ensure proper signalling regulations during organogenesis [2,3,4].

Among the many transcription regulators, FUSE (far upstream element) binding proteins (FBPs) are regarded as important molecular tools during gene expression regulation [10]. One of these genes, *Fubp1*, expressed in developing tooth germ, is an evolutionary conserved gene that is present in eukaryotes and regulates c-Myc [11,12], a transcription factor involved in the regulation of gene expression [13,14,15]. As a c-Myc promoter, *Fubp1* binds to FUSE, serving as a sensor of transcriptional activity [16]. In a range of tissues, *Fubp1* has been reported to be involved in transcription, translation and post-translational regulation of genes [12]. Several studies reported the abnormal expression of *Fubp1* in malignant cell lines with altered cellular physiology including proliferation, migration and apoptosis [17,18]. Therefore, it is necessary to emphasize the importance of *Fubp1* in cellular behaviors, particularly cell cycle regulation and differentiation, the most important processes for the proper structural formation of organs. However, its precise expression pattern and developmental roles in odontogenic tissue are not yet understood.

During odontogenesis, the spatio-temporal regulation of genes at cap and bell stages play crucial roles for functional and structural formation of tooth [6]. In this study, the functional evaluation of *Fubp1*, a transcriptional regulator gene, is performed at cap stage through siRNA knock-down and in vitro organ cultivation system to evaluate the developmental role of *Fubp1* prior to differentiation of odontoblasts and ameloblasts. Copious reports have revealed that complex interactions of signalling molecules at cap stage with precise transcriptional regulations would result in tooth morphogenesis, especially with hard tissue matrix formation at the bell stage [6,7,8,9]. In this study, as a pioneer work, we attempted to examine the detailed function of *Fubp1* in tooth development at the cap stage, which shows dynamic differentiation sequences with a well-studied signalling network for tooth morphogenesis. Since the functional evaluation of transcriptional regulators, even though their importance in development, was difficult to perform in knockout animal models [3,6,9,19], this in vitro organ cultivation approach would be the plausible model system to provide the fine-tuning of transcription regulations by *Fubp1*.

## 2. Results

### 2.1. Expression Pattern and Functional Evaluation of Fubp1

The developing tooth was at cap and bell stages of tooth development (Figure 1a–c). During the cap stage (Figure 1d,e), *Fubp1* was broadly expressed in the enamel epithelium and condensed mesenchyme, with intense expression in the cervical loop (Figure 1d–f). Distinct expression of *Fubp1* was observed in the early bell stage (Figure 1f) along the inner enamel epithelium (IEE) and dental papilla (DP), which are presumptive ameloblast- and odontoblast-forming cells, respectively. Its expression was also observed along the outer enamel epithelium (OEE) and stratum intermedium at E16 (Figure 1f). There was 52% down regulation of *Fubp1* by siRNA during in vitro organ cultivation (Figure 2a) and the developing teeth remained at bell stage after cultivation at E14 for two days (Figure 2b,c). In order to define the cellular physiology regulated by this gene, we examined the precise localization patterns of c-Myc (transcription regulator) and Ki67 (cell proliferation marker) through immunohistochemistry (Figure 2). Knocking down *Fubp1* at the cap stage of tooth development resulted into a disrupted patterned arrangement of c-Myc positive cells in the IEE and OEE (Figure 2d,f). Particularly, a greater number of c-Myc positive cells are observed in the IEE, whereas less are observed in the OEE of the *Fubp1* knock-down specimen (Figure 2d,f,i Appendix A). Similar to these reactions, Ki67 reactions showed more ki67 positive cells along the IEE between secondary enamel knot-forming regions and decreased positive reactions in the OEE (Figure 2e,g; Appendix A). The mesenchymal cells adjacent to the secondary enamel knot of control specimens showed more proliferative cells compared with the same region of *Fubp1* knock-down (Figure 2e,g). In contrast, there was not much change in the c-Myc protein expression level after knock-down of *Fubp1* when examined by Western blot analysis (Appendix A). In addition, epithelial cell rearrangement, one of the important cellular events for epithelial morphogenesis, was examined using immunostaining against E-cadherin and ROCK1, and phalloidin staining, as were examined in previous study [20]. The localization of E-cadherin in epithelial cells revealed that cellular adhesion seemed weaker in the inner enamel epithelial cells of the knock-down specimen compared with the control (Figure 3a,d). Similarly, compared with the *Fubp1* knock-down specimen, strong localization of ROCK1 and phalloidin staining were observed along the basement membrane of IEE and the adjacent cells of the DP in the control specimens (Figure 3b,c,e,f).

### 2.2. Altered Expression Patterns of Signalling Molecules by Fubp1 Knock-Down

Tooth development is regulated with the complex interactions of various signalling molecules [6,21], and therefore most of the tooth development-related signalling molecules were examined using qPCR to evaluate the knock-down effects of *Fubp1*. After knocking down *Fubp1* expression at E14 for 24 h, signalling molecules, including *Bmp2*, *Bmp4*, and *Rock1*, were found to be downregulated (Figure 4a). Conversely, ß-*catenin*, *Fgf4*, *Shh*, and *Lef1* were up-regulated, suggesting the modulating role of *Fubp1* during the cap stage of tooth development. Moreover, enamel- and dentin-specific markers, such as *Amelx*, *Ambn*, *Dspp,* and *Dmp1*, were found to be down-regulated following *Fubp1* knock-down (Figure 4a). After observing the expression patterns of tooth-related signalling molecules, in situ hybridization of sections was performed considering *Bmp4* as a differentiation factor and *Fgf4* as a proliferative factor during tooth development. The expression of *Fgf4* was continuous along the enamel rope of tooth with *Fubp1* knock-down (Figure 4c), whereas, its expression was restricted to the enamel knot in the control (Figure 4b). In contrast, the expression of *Bmp4* was reduced in the mesenchyme of the specimen with *Fubp1* knock-down (Figure 4e) compared with the control (Figure 4d).

### 2.3. Fubp1 in Tooth Morphogenesis

We employed renal capsule transplantation and examined the morphological alterations of the tooth to define the knock-down effect of *Fubp1* at the cap stage of tooth development, as were examined in previous report [21]. The arrangement of the dentin matrix along the cusp of the *Fubp1* knock-down tooth was abrogated in 1-week renal capsule calcified tooth when compared with the same region of the control specimen (Figure 5a,d). In addition, the *Fubp1* knock-down tooth showed relatively weaker expressions of *Dspp* and *Amelx* mRNA along the cusp area when compared with the equivalent region of the control (Figure 5b,c,e,f). The expression pattern was also reduced to a smaller area in *Fubp1* knock-down specimen compared to control (Figure 5b,c,e,f). To check whether the reduced expression pattern of *Dspp* and *Amelx* corresponds with the NESTIN localization, we performed immunostaining of NESTIN, an odontoblast differentiation marker, and the result showed the abrogated and weaker localization of NESTIN in the secreting odontoblasts and its processes relative to the control (Appendix A). Furthermore, after observing relatively weaker expression of Amelx and Dspp in 1-week *Fubp1* knock-down renal calcified tooth, and therefore, in order to evaluate the long term effect of *Fubp1* knock-down, 3-week kidney capsule transplantation was employed and the thickness of dentin matrix and cuspal patterning were examined (Figure 5g–l). Although there were no significant differences in the overall dentin thickness between the control and knock-down specimen, the *Fubp1* knock-down exhibited altered dentin and enamel patterned arrangements. SEM indicated an altered pattern of the enamel rod and interrod in the *Fubp1* knock-down tooth compared with the control (Figure 5h,k). Moreover, the control specimen displayed well-arranged dentinal tubules (Figure 5i), but the arrangement of dentinal tubules was abrogated, especially in the cusp area of the *Fubp1* knock-down tooth (Figure 5l). Furthermore, 3-week renal calcified tooth showed alteration in crown height in the *Fubp1* knock-down specimen (Appendix A). Comparatively shorter crown heights were seen in the *Fubp1* knock-down specimen compared with the control. However, there was no variation in cusp height.

## 3. Discussion

Copious studies with experimental models, such as cell lines and animals models, have already been undertaken to identify the principle mechanisms of tooth development [22], however, there is still poor understanding of the key modulations among the candidate molecules in organogenesis processes. In this study, we firstly examined the precise expression pattern of a eukaryotic transcription regulator gene, *Fubp1*, along the epithelium and mesenchyme of the developing tooth (Figure 1d–f). Our results showed the co-expression pattern with Hoxc12, zfp36I1, Taf10, Cnbp and Ehmt2 [23] as candidate molecules involved in directing the tooth morphogenesis via modulating the signalling molecules during embryogenesis. Particularly, in this study, as the *Fubp1* knock-out mouse is embryonically lethal [19], we tried to evaluate its crucial functional roles through transient *Fubp1* knock-down and an in vitro organ cultivation system. Previous reports mention that c-Myc, a eukaryotic transcription regulator, is involved in cellular proliferation and differentiation [24,25,26]. It is also known that *Fubp1* regulates the cell cycle progression through both c-Myc- dependent [27,28] and -independent [29] mechanisms. Additionally, cellular events such as proliferation and differentiation are very important during organogenesis. Our study showed that knocking down *Fubp1* at E14 for 2 days increased cellular proliferation along the IEE between the secondary enamel knot-forming region (Figure 2e,h), indicating that the IEE cells are still proliferating, whereas the control cells are already set for differentiation. The rate of cellular proliferation usually decreases as cells differentiate [30], and so it seems that *Fubp1* enhances the differentiation of the IEE specifically at the cusp-forming region. The region-specific cellular proliferation after *Fubp1* knock-down draws particular attention during in vitro organ cultivation. Increased proliferating cells along the IEE and decreased proliferating cells along the OEE in *Fubp1* knock-down tooth (Figure 2) implied that the modulating role of *Fubp1* is important along the entire enamel epithelium. We hypothesized that these altered cell proliferation patterns would result from the altered localization pattern of c-Myc (Figure 2d,g). However, *Fubp1* knock-down tissue did not show any obvious change in the c-Myc protein expression when examined by Western blot (Appendix A), in agreement with a previous report [29]. Interestingly, immunohistochemistry analysis revealed a disrupted patterned arrangement of positive reactions against c-Myc in the IEE and OEE (Figure 2d,g). These results suggest that knock-down of *Fubp1*, one of the potential regulators of c-Myc, [27,28], would result altered tooth morphogenesis through changed cellular events.

The spatio-temporal expression of signalling molecules is critical for proper morphogenesis of the tooth and the knock-down of *Fubp1* at the cap stage alters the expression of signalling molecules, including *Bmp, Fgf4, Shh, Dspp, B-catenin* and *Rock1* (Figure 4a). The mesenchymal expression of Bmp signalling during the bell stage of tooth development is crucial for the signalling interaction between odontoblasts and ameloblasts [31]. Knock-out of *Bmp2* and *Bmp4* caused a low odontoblast differentiation marker expression, with reduced dentin thickness [32,33,34]. Knock-down of *Fubp1* led to an observed decrease in the expression of *Bmp4* mRNA, indicating its role in *Bmp* signalling (Figure 4d,e). At the bell stage of tooth development, inner enamel epithelial cells start to differentiate into ameloblasts and DP cells differentiate into odontoblasts [31] and the expression pattern of *Fgf4* plays a crucial role in future cusp patterning [35,36]. We observed a continuous expression pattern of *Fgf4* along the enamel rope in the *Fubp1* knock-down tooth (Figure 4b,c). Consistent with this result, it has been reported that its expression is restricted along the enamel knot during the bell stage of mice molar development [21,35,37]. Copious studies have reported that various paracrine factors are involved in tooth morphogenesis, including Bmps, Fgfs, Shh, and Wnts [3,6,7,8,9]. Among these signalling molecules, Fgfs and Shh are well known enamel knot related signalling molecules and putatively play important roles in odontoblast differentiation and tooth morphogenesis [3,6,9]. In our study, three-week *Fubp1* knock-down renal calcified teeth showed the altered crown height (Appendix A) and based on these morphological alterations, we decided to examine expression patterns of signalling molecules which would be involved in the morphological alterations, particularly with *Bmp2*, *Bmp4*, and *Fgf4* [3,6,9] (Figure 4). The knock-down effect of *Fubp1* up-regulated *Fgf4* and altered the crown height (Figure 4, Appendix A). However, a similar expression pattern of *Fgf4* resulted in a long cusp height in gerbils [35] suggesting that *Fgf* signalling not only to cuspal height morphogenesis but also with crown height. Our study showed down-regulation of *Bmp2* and *Bmp4* after *Fubp1* knock-down during the bell stage. This result suggests that, in order to rescue weak *Bmp* signalling from the mesenchyme, *Fgf4* would be up-regulated during reciprocal interaction between the epithelium and mesenchyme as in a previous report [38]. In addition, the increased expression of *Shh* in the *Fubp1* knock-down could alter crown height because *Shh* signalling plays a crucial role during cytodifferentiation of ameloblasts and odontoblasts, and for cusp formation [39]. Meanwhile, the cellular adhesion and actin rearrangement play a crucial role during differentiation of ameloblasts [40]. In this study, the weak localization of ROCK1 and phalloidin staining along the IEE of the *Fubp1* knock-down tooth implied that *Fubp1* would modulate actin rearrangement during differentiation of ameloblasts before the formation of cusp morphogenesis because ROCK contributes to maintaining ameloblast polarity and enamel matrix secretion [40].

At the secretory stage of mice molar development, *Fubp1* expression was observed along the secretory ameloblasts and odontoblasts (Figure 5b,c,e,f), similar to the expression pattern of *Amelx* and *Dspp* [41]. As the co-expression patterns of genes imply their shared functions in development and disease prediction [42], it showed that *Fubp1,* co-expressed with *Amelx* in ameloblasts and *Dspp* in odontoblasts, has a modulating role during enamel and dentin matrix secretion (Figure 5). Odontoblasts and ameloblasts initiate secretion of the extracellular matrix proteins from the late bell stage, and the knock-down of enamel- and dentin-related genes at this stage could alter the secretory function of cells. The conditional deletion of enamel- and dentin-specific genes, such as *Amelx*, *Ambn*, *Enam,* and *Dspp,* alters the entire phenotype of the tooth enamel and dentin [43,44,45]. Besides, other paracrine and transcription factors also play a role in the secretory function of cells. Our results suggest that the transient knock-down of *Fubp1* at cap stage (E14) might alter the proliferation and differentiation of pre-ameloblasts and pre-odontoblasts (E16) (Figure 2) prior to differentiation of mature ameloblasts and odontoblasts at late bell stage (E18.5). This alteration in cellular events resulted comparatively reduced dentin and enamel matrix in *Fubp1* knock-down renal calcified tooth (Figure 5a,d). In addition, the weak expression of *Amelx* and *Dspp* mRNA, and abrogated localization of NESTIN in 1-week renal calcified tooth implied that *Fubp1* is involved in amelogenesis and dentinogenesis during tooth matrix secretion (Figure 5, Appendix A). The synergistic role of the IEE and OEE is crucial for the Hertwig epithelial root sheath (HERS) development, and therefore, the retarded proliferation and c-Myc localization along the OEE of *Fubp1* knock-down tooth (Figure 2) suggested that *Fubp1* would modulate the tooth morphogenesis by controlling harmonious cellular events. To further elucidate its role, the morphology of the three-week renal calcified tooth was observed (Appendix A) as in previous reports [39]. It showed that knocking down *Fubp1* altered the enamel rod- interrod architecture and abrogated dentinal tubules (Figure 5), suggesting that *Fubp1* regulates the transcription of dentin- and enamel-forming genes during the secretory stage of molar development. Moreover, the short crown height in the *Fubp1* knock-down tooth might be explained by abrogation of cellular physiology, including proliferation and adhesion, during the bell stage of tooth development (Figure 2 and Figure 3). E-cadherin enhances proliferation and plays a major role in tissue morphogenesis [46,47]. However, it was weakly localized along the IEE in the *Fubp1* knock-down tooth (Figure 3) which would cause altered tooth morphogenesis (Appendix A).

Dental hard tissues, especially dentin and enamel, are required to withstand the mechanical stresses during mastication. Therefore, their precise patterned arrangement for relieving the forces is crucial for the quality and longevity of teeth. Especially during enamel and dentin structural formation, the patterned array would result from balanced cellular events of dental cells, including proliferation and differentiation, in a spatio-temporal manner. However, the knock-down of *Fubp1* disrupted the patterned array of those dental tissues through alterations in signalling events including proliferation and adhesion (Figure 2 and Figure 3 and Appendix A). To present this, only paracrine and transcription factors were paid attention during tooth development, especially for matrices formation [6,7]. Although substantial knowledge is accumulating about these signalling molecules, we still do not understand the fundamental mechanisms of tooth development. To overcome this knowledge gap, it is necessary to examine the developmental function of modulators which are known to have the integrated function of signalling molecules including transcription and paracrine factors during maintenance of homeostasis. Until now, the in vitro organ cultivation with functional analysis methods would be the only way to analyze the precise function of complicated signalling molecules, which showed lethality during the embryonic stage when genetically manipulated. In the near future, it will be vital to discover the fundamental and integrative signalling network of such genes on tooth development in order to apply these signalling pathways in regenerative medicine fields for practical purpose.

## 4. Material and Methods

All experiments involving animals were performed according to the guidelines of the Kyungpook National University, School of Dentistry, Intramural Animal Use and Care Committee (KNU-2020-0107).

### 4.1. Animals

ICR mouse embryos were obtained from time-mated pregnant mice that were maintained in an optimal environment. The day the vaginal plug was confirmed was designated embryonic day 0 (E0). All experimental protocols were approved (2020/07/23) by the Kyungpook National University, School of Dentistry, Animal Care and Use Committee (KNU-2020-0107) and followed the ARRIVE guidelines for the care and use of laboratory animal [48]. All experiments described in this study were performed 3 or more times independently.

### 4.2. In Situ Hybridization

In-situ hybridization of sections was performed at 68 °C using digoxigenin (DIG)-labelled RNA probes: *Amelx* (490 bp), *Dspp* (934 bp), *Bmp4* (1000 bp), *Fgf4* (620 bp), and *Fubp1* (419 bp). Probes were hybridized overnight by following standard protocols [37].

### 4.3. siRNA Transfection, In Vitro Organ Cultivation and Kidney Transplantation

The embryonic mice molar tooth germs at E14 were cultivated for 1 and 2 days, and transplanted into the kidney capsule, as described by [37,49]. During cultivation, embryonic tooth specimens were transfected with scrambled siRNA (negative control) (Cat# SR30004, Origene Technologies Inc., Rockville, MD, USA), *Fubp1* siRNA (Cat# SR418703, Origene Technologies Inc., Rockville, MD, USA) using siTran transfection reagent (Cat# TT30001, Origene Technologies Inc., Rockville, MD, USA), in separate culture dishes for 1 day in Opti-MEM (Cat# 31985-070, Gibco, Grand Island, NY, USA). The siRNAs were used at a final concentration of 100 nM. The naive and scrambled control did not show any differences in the expression of the gene (Data not shown).

### 4.4. Histology, Immunohistochemistry and Western Blotting

Histology and immunostaining were carried out, as described previously [20]. Hematoxylin and eosin (H&E) and Masson’s trichrome (MTC) staining were employed to examine the detailed morphological changes [50]. Primary antibodies were directed against Ki67 (Cat# RM-9106-s, Thermo Scientific, Waltham, MA, USA), NESTIN (Cat# ab11306, Abcam, Cambridge, MA, USA), c-Myc (Cat# ab32072, Abcam), E-cadherin (Cat# AF748, R&D Systems, Minneapolis, MN, USA) and ROCK1 (Cat# ab45171, Abcam). The secondary antibodies used in this study were biotinylated goat anti-rabbit or anti-mouse IgG. Immunocomplexes were visualized using a diaminobenzidine tetrahydrochloride (DAB) reagent kit (Cat# C09-12; C09-100; GBI Labs, Bothell, WA, USA) and goat anti-rabbit IgG Flamma 488 (Cat# RSA1241, BioActs, Incheon, Korea). Alpha-tubulin (G436) and AC-74 were used for Western blot experiments.

### 4.5. Quantitative PCR (qPCR)

Quantitative PCR was performed as described previously [21]. RNA was extracted from E14 + 1 day cultivated tooth germ, and qPCR was employed to examine the knock-down effect of *Fubp1*. The qPCR results for each sample were normalized to those of Hprt and expressed as normalized ratios. Data were represented as means ± standard deviations. The mean expression levels were compared between the experimental and control groups using the Student’s *t*-test at *p* < 0.05 level of significance. The primers used in this study are presented in Table 1.

### 4.6. Ground Section and Scanning Electron Microscopy (SEM)

Ground sections of teeth were prepared as described by [21,37]. Teeth were stained with Villanueva stain for 3 days, followed by dehydration, infiltration with a mixture of propylene oxide and then embedded in resin at 60 °C for 5 days. Sections were prepared by using a diamond saw (Accutom 50, Ballerup, Denmark), ground (Rotopol-35, Ballerup, Denmark), then mounted on slides in historesin mounting media (Leica, Hanau, Germany). SEM (SEM; JEOL JSM-6700 F Field emission SEM, Tokyo, Japan) was performed, as detailed by [35].

## Figures and Tables

**Figure 1 ijms-21-08079-f001:**
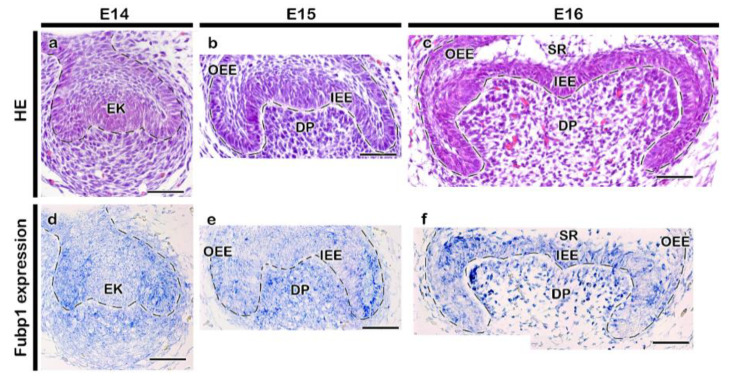
Expression of *Fubp1* in developing tooth. H&E staining of developing tooth (**a**–**c**) and section in situ hybridization of *Fubp1* at E14-E16 (**d**–**f**). At E14 and E15, *Fubp1* is expressed broadly at epithelium and mesenchyme. At E16, *Fubp1* is expressed in IEE and DP, however, more intense expression is observed in IEE. The dotted lines demarcate the boundary of tooth epithelium (**a**–**f**). EK, enamel knot; IEE, inner enamel epithelium; OEE, outer enamel epithelium; SR, stellate reticulum; DP, dental papilla. Scale bars: 50 μm (**a**–**f**).

**Figure 2 ijms-21-08079-f002:**
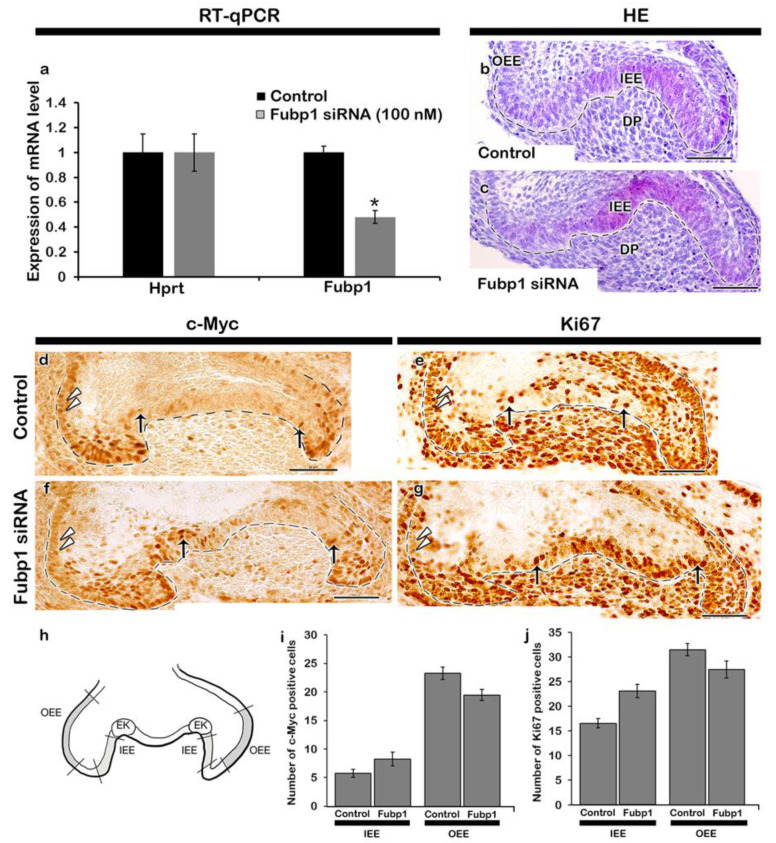
Altered c-Myc localization and cellular proliferation after *Fubp1* knock-down. *Fubp1* siRNA (100 nM) downregulates the expression of *Fubp1* during in vitro organ cultivation at E14 for 1 day as determined by quantitative PCR (**a**). Hematoxylin and eosin staining of E14 + 2 control and *Fubp1* siRNA-transfected tooth germ showing IEE and DP (**b**,**c**). Immunohistochemistry of c-Myc and Ki67 showing increased number of c-Myc and ki67 positive cells at IEE and decreased number of c-Myc and Ki67 cells at OEE in the *Fubp1* knock-down specimen compared to control (**d**–**g**). Schematic diagram showing region of interest for cell count (**h**). Statistical analysis showing increased and decreased c-Myc and Ki67 positive cell numbers along the IEE and OEE respectively in the *Fubp1* knock-down specimen (**i**,**j**). Arrows and arrowheads indicate c-Myc positive cells (**d**,**f**) and Ki67 positive cells (**e**,**g**). EK; enamel knot, IEE; inner enamel epithelium, OEE; outer enamel epithelium, DP; dental Papilla. Scale bars: 50 μm (**b**–**g**).

**Figure 3 ijms-21-08079-f003:**
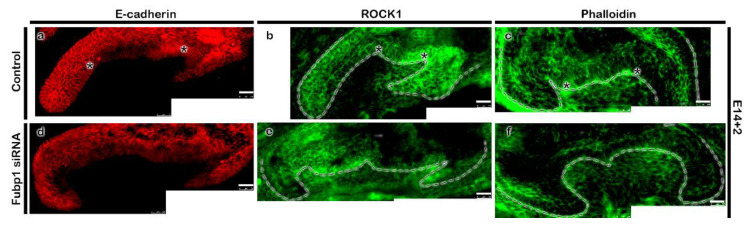
Altered epithelial cell rearrangements in *Fubp1* knock-down molar. Knocking down *Fubp1* at E14 for 2 days shows decreased localization of E-cadherin and ROCK1 along the inner enamel epithelium (**a**–**e**). Similarly, weak phalloidin staining is observed in the basement membrane of inner enamel epithelium along the secondary enamel-knot in the *Fubp1* knock-down tooth compared with control (**c**,**f**). The dotted lines demarcate the boundary of tooth epithelium (**a**–**f**).* in control indicates comparatively increased localization of respective proteins along inner enamel epithelium when compared with the same region of *Fubp1* knock-down tooth (**a**–**f**). Scale bars: 20 μm (**a**–**f**).

**Figure 4 ijms-21-08079-f004:**
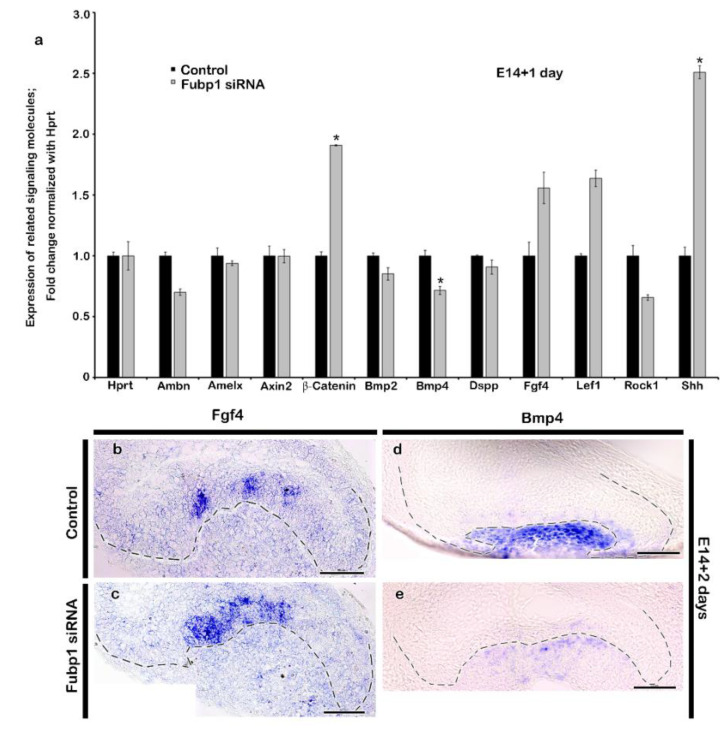
Expression patterns of tooth related signalling molecules after *Fubp1* knock-down. Knock-down of *Fubp1* results altered expression of tooth development related signalling molecules including *Bmp*, *Wnt*, *Shh* and *Fgf4* as detected by qPCR (**a**) and section in situ hybridization (**b**–**e**). Especially, *Bmp2*, *Bmp4* and *Rock1* are downregulated, whereas β-catenin, *Fgf4*, *Lef1* and *Shh* are upregulated. Enamel- and dentin-specific signalling molecules: *Ambn*, *Amelx* and *Dspp* are downregulated in *Fubp1* knock-down tooth. Similarly, section in situ hybridization shows increased expression of *Fgf4* and decreased expression of *Bmp4* along enamel and mesenchyme of the *Fubp1* siRNA transfected tooth germ respectively compared with the control (**b**–**e**). * indicates *p* < 0.05. Scale bars: 50 μm (**b**–**e**).

**Figure 5 ijms-21-08079-f005:**
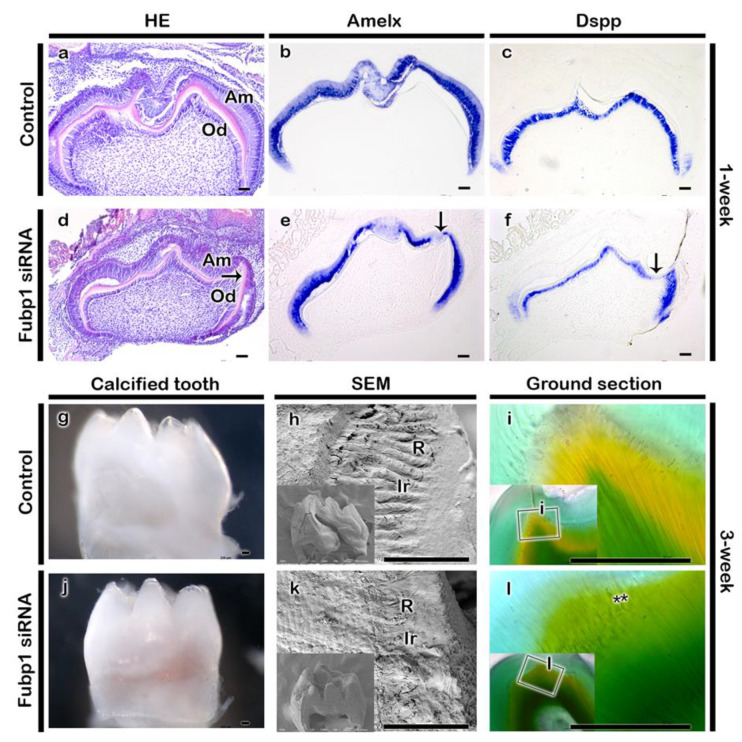
Renal capsule transplanted teeth for one-week and three-week. One-week (**a**–**f**) and three-week (**g**–**l**) renal calcified tooth. H&E staining showing ameloblast and odontoblast layer with enamel and dentin matrix in one-week calcified tooth. (**a**,**d**). Amelx and Dspp expression is relatively weaker in the *Fubp1* knock-down tooth compared with the control (**b**,**c**,**e**,**f**). Dspp and Amelx mRNA expression is comparatively weaker in knock-down tooth especially along cusp region (arrows, (**e**–**f**)). Three-week renal calcified control and *Fubp1* knock-down tooth (**g**,**j**). SEM showing patterned enamel rod and interrod in the control specimen (**h**) which is abrogated in the knock-down tooth (**k**). Resin section shows well organized dentinal tubules in the control specimen (**i**), while dentinal tubules are not well arranged (indicated by double astericks) especially along the cusp of thee *Fubp1* knock-down tooth (**l**). Square box indicates the magnified view (**i**–**l**). SEM, scanning electron micrograph; HE, hematoxylin and eosin staining; R, rod; Ir, interrod; Amelx, amelogenin; Dspp, dentin sialophosphoprotein; De, dentin; Am, Ameloblast; Od, odontoblast. Arrow in ‘d’ indicates comparatively thinner dentin layer in the *Fubp1* knock-down tooth. Scale bars: 50 μm (**a**–**l**).

**Table 1 ijms-21-08079-t001:** List of primers used in the study.

Gene	Accession	Primer Sequence		Product Size (bp)	Remarks
Fubp1	NM_001355372.1	Forward	GCATCAGCAGCAAAGCAGAT	419	Probe synthesis
		Reverse	CAGCACCAGTGTTTTGAGGC		
		Forward	TGCAGCAAAAATTGGGGGTG	173	qPCR
		Reverse	ATCTGCTTTGCTGCTGATGC		
Ambn	NM_001303431.1	Forward	TTCTTGCTTTCCCCAATGAC	234	Amelogenesis
		Reverse	GGTGCACTTTGTTTCCAGGT		
Amelx	XM_017348358.1	Forward	GCAGCCGTATCCTTCCTATGGTT	120	Amelogenesis
		Reverse	GGAAGGTGGTGATGAGGCTGAA		
Axin2	BC057338.1	Forward	TGAAGAAGAGGAGTGGACGT	115	Wnt signalling
		Reverse	AGCTGTTTCCGTGGATCTCA		
β-catenin	NM_007614.3	Forward	TGACCTGATGGAGTTGGACA	104	Wnt signalling
		Reverse	TGGCACCAGAATGGATTCCA		
Bmp2	NM_007553.3	Forward	AAGTGGCCCATTTAGAGGAG	104	Bmp signalling
		Reverse	CCATGGCCTTATCTGTGACC		
Bmp4	NM_007554.3	Forward	ACCTCAAGGGAGTGGAGATT	113	Bmp signalling
		Reverse	GATGCTTGGGACTACGTTTG		
Dspp	NM_010080.3	Forward	GTGGGGTTGCTACACATGAAAC	169	Dentinogenesis
		Reverse	CCATCACCAGAGCCTGTATCTTC		
Fgf4	NM_010202.5	Forward	TCGCCTACCATGAAGGTAAC	114	Fgf signalling
		Reverse	TCTCCATCGAGAGAAAGTGC		
Lef1	NM_010703.4	Forward	ACAGCGACGAGCACTTTTCT	82	Shh Signalling
		Reverse	TGTCTGGACATGCCTTGCTT		
Rock1	NM_009071.2	Forward	GGTATCGTCACAAGTAGCAGCATC	140	Wnt signalling
		Reverse	TAAACCAGGGCATCCAATCCA		
Shh	NM_009170.3	Forward	CCAAAGCTCACATCCACTGT	131	Shh Signalling
		Reverse	GGGACGTAAGTCCTTCACCA		
Hprt	NM_013556.1	Forward	CCTAAGATGATCGCAAGTTG	86	Internal standard
		Reverse	CCACAGGGACTAGAACACCTGCTAA		

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
