# Peer review of "Developmental Roles of FUSE Binding Protein 1 (Fubp1) in Tooth Morphogenesis"

_ijms, 2020, doi:10.3390/ijms21218079_

Round 1
Reviewer 1 Report
The article is written quite well, generally speaking, with some minor problems to be corrected/addressed. The study itself is important for the understanding of molecular mechanisms underlying tooth development and correct differentiation. I like that the authors made sure that the pictures are quite clear and neat, so it is easy even for the "unspecialised eye" to understand what the authors refer to. I believe that if authors respond accordingly to my remarks and alter the manuscript accordingly, it should be published.
general remarks:
1. M&M: there is a missing section about siRNA. Please add.
2. What is your control? I presume these are just normal "naive" explants of tooth buds. However, it is a good practice that when using siRNA technique, you have performed the control siRNA study (i.e. using any foreign siRNA, like against GFP etc., that theoretically should have no effect whatsoever on the global expression of any genes), to definitely exclude non-specific effects of it on the investigated process. I believe that this is by far the most important part that was omitted/not indicated and it should be added to the manuscript before it is considered for publication.
3. the siRNA-led knockdown of Fubp1: authors refer to it as "effective". One could argue that 52% downregulation of the gene expression is not very effective, when siRNA technique normally is considered effective at around 70-80% knockdown. Whereas I agree that 52% may be enough to cause differences, I would re-think calling it "effective".
I noticed that the stages of tooth development are referred to only in Discussion. I believe that for the clarity, this should be put in the Introduction, so the Reader can easier follow the presented results.
minor remarks:
the scale bars on the pictures should perhaps represent the unified scale. F.ex, take 50um as a point of reference and place the scale bars that will correspond with that size on the pictures, rather than keeping bars neatly the same size on the pics, and then explaining in the legend that sometimes it means 10um, sometimes 50um etc.
some sentences need to be rewritten to make them clearer
Please see the attached file with comments for the rest of minor issues highlighted

Author Response
The article is written quite well, generally speaking, with some minor problems to be corrected/addressed. The study itself is important for the understanding of molecular mechanisms underlying tooth development and correct differentiation. I like that the authors made sure that the pictures are quite clear and neat, so it is easy even for the "unspecialised eye" to understand what the authors refer to. I believe that if authors respond accordingly to my remarks and alter the manuscript accordingly, it should be published.
General remarks:
- M&M: there is a missing section about siRNA. Please add.
: As reviewer pointed out, we carefully added the detailed methods for siRNA transfection in the material and methods section and we also more discussed about its transfection protocol.
Page 9: Line 303-310
siRNA transfection, in vitro organ cultivation and kidney transplantation
The embryonic mice molar tooth germs at E14 were cultivated for 1 and 2 days, and transplanted into the kidney capsule, as described by [35,48]. During cultivation, embryonic tooth specimens were transfected with scrambled siRNA (negative control) (Cat# SR30004, Origene Technologies Inc., USA), Fubp1 siRNA (Cat# SR418703, Origene Technologies Inc., USA) using siTran transfection reagent (Cat# TT30001, Origene Technologies Inc., USA), in separate culture dishes for 1 day in Opti-MEM (Cat# 31985-070, Gibco, USA). The siRNAs were used at a final concentration of 100 nM. The naive and scrambled control did not show any differences in the expression of the gene (Data not shown).
- What is your control? I presume these are just normal "naive" explants of tooth buds. However, it is a good practice that when using siRNA technique, you have performed the control siRNA study (i.e. using any foreign siRNA, like against GFP etc., that theoretically should have no effect whatsoever on the global expression of any genes), to definitely exclude non-specific effects of it on the investigated process. I believe that this is by far the most important part that was omitted/not indicated and it should be added to the manuscript before it is considered for publication.
: As reviewer inquired, the control used in this study means “the scrambled control”. During cultivation, embryonic tooth specimens were transfected with scrambled control, which we wrote as ‘Control’ in the manuscript. The naive and scrambled control did not show differences in the histogenesis and expression patterns of the genes. The details of scrambled control and Fubp1 siRNA were prepared in the materials and methods section.
Page 9: Line 303-310
siRNA transfection, in vitro organ cultivation and kidney transplantation
The embryonic mice molar tooth germs at E14 were cultivated for 1 and 2 days, and transplanted into the kidney capsule, as described by [35,48]. During cultivation, embryonic tooth specimens were transfected with scrambled siRNA (negative control) (Cat# SR30004, Origene Technologies Inc., USA), Fubp1 siRNA (Cat# SR418703, Origene Technologies Inc., USA) using siTran transfection reagent (Cat# TT30001, Origene Technologies Inc., USA), in separate culture dishes for 1 day in Opti-MEM (Cat# 31985-070, Gibco, USA). The siRNAs were used at a final concentration of 100 nM. The naive and scrambled control did not show differences in histogenesis and the expression patterns of the genes (data not shown).
- The siRNA-led knockdown of Fubp1: authors refer to it as "effective". One could argue that 52% downregulation of the gene expression is not very effective, when siRNA technique normally is considered effective at around 70-80% knockdown. Whereas I agree that 52% may be enough to cause differences, I would re-think calling it "effective".
: As reviewer suggested, we omitted the term “effective”, and have prepared the sentence without misleading the meaning of knockdown using siRNA as: “There was 52% down regulation of Fubp1 by siRNA during in vitro organ cultivation (Fig. 2a)”.
Page 2: Line 82-83
There was 52% down regulation of Fubp1 by siRNA during in vitro organ cultivation (Fig. 2a) and the developing teeth remained at bell stage after cultivation at E14 for 2 days (Fig. 2b, c).
- I noticed that the stages of tooth development are referred to only in Discussion. I believe that for the clarity, this should be put in the Introduction, so the Reader can easier follow the presented results.
: As reviewer suggested, we have added the information of mouse tooth developmental stages including embryonic stages E13, E14 and E16, which represent bud, cap and bell stage respectively. As reviewer knows well, we did siRNA transfection in cap stage and we checked the expression pattern of Fubp1 in cap and bell stages. Also we prepared the proper references for better understanding the tooth development itself.
Page 2: Line 47-49
Following epithelial invaginations, through the combined interactions of signalling molecules and transcription factors, morphogenesis proceeds with bud, cap and bell stages at mouse embryonic stages E13, 14 and 16 respectively [6,7].
Minor remarks:
- The scale bars on the pictures should perhaps represent the unified scale. F.ex, take 50um as a point of reference and place the scale bars that will correspond with that size on the pictures, rather than keeping bars neatly the same size on the pics, and then explaining in the legend that sometimes it means 10um, sometimes 50um etc.
: As reviewer suggested, we prepared uniformity (50 µm) in the scale bar, especially in the Figure 5.
- Some sentences need to be rewritten to make them clearer. Please see the attached file with comments for the rest of minor issues highlighted
: We rewrote all the sentences in the manuscript, as reviewer suggested, with careful for better understanding the contents.
Page 2: Line 82-83
There was 52% down regulation of Fubp1 by siRNA during in vitro organ cultivation (Fig. 2a) and the developing teeth remained at bell stage after cultivation at E14 for 2 days (Fig. 2b, c)
Page 6: Line 163-166
The main extracellular proteins of enamel: Amelx and Dspp, were weakly expressed in Fubp1 knock-down tooth, and therefore, in order to evaluate the long term effect of Fubp1 knock-down, 3-week kidney capsule transplantation was employed and the thickness of dentin matrix and cuspal patterning were examined (Fig. 5g–l).
Page 7: Line 198-200
Particularly, in this study, as the Fubp1 knock-out mouse is embryonically lethal [23], we tried to evaluate its crucial functional roles through transient Fubp1 knock-down and using an in vitro organ cultivation system.
Page 7: Line 203-204
Additionally, cellular events such as proliferation and differentiation are very important during organogenesis.
Page 7: Line 210-212
Increased proliferating cells along the IEE and decreased proliferating cells along the OEE in Fubp1 knock-down tooth (Fig. 2) implied that the modulating role of Fubp1 is important along the entire enamel epithelium.
Page 7: Line 214-216
However, Fubp1 knock-down tissue did not show any obvious change in the c-Myc protein expression when examined by Western blot (Supplementary Figure S3), in agreement with previous report [27].
Page 7: Line 232-234 to Page 8: Line 235
The knock-down effect of Fubp1 up-regulated Fgf4 and altered the crown height (Fig. 4, Supplementary Figure. S4), however, similar expression pattern of Fgf4 resulted into long cusp height in gerbils [35] suggesting that Fgf signalling not only to cuspal height morphogenesis but also with crown height.
Page 9: Line 322
Quantitative PCR was performed as described previously [20].
: As reviewer suggested, we newly prepared Figure 1e (Expression of Fubp1 in E15 tooth germ) for better understanding.
Reviewer 2 Report
This study aims to analyze the developmental function of FUSE binding protein 1 (Fubp1) gene during murine tooth morphogenesis. The authors used gene expression analyses (RT-qPCR, in situ hybridization), an in vitro loss-of-function tooth organ model and renal transplantation. They found that Fubp1 was expressed in the enamel organ and condensed mesenchyme during normal tooth formation. When the Fubp1 gene was knocked down, there were changes in the expression of tooth development related gene such as Bmps and Fgf4, localization patterns of c-Myc and cell proliferation in epithelium and mesenchyme, related with altered tooth morphogenesis, decreased amelogenin and dentin sialophosphoprotein expressions and disrupted enamel rod and interrod formation in 1- and 3-week renal transplanted teeth respectively. The authors suggest that Fubp1 modulates cell physiology of dentinogenesis and amelogenesis by regulating expression pattern of signalling molecules during mice molar development to achieve the proper structural formation of hard tissue matrices and crown morphogenesis.
Strengths: The expression analyses appear to be well done, with well executed gene expression analyses. The use of an organ transplant, knockdown technology and kidney transplantation also provided additional data to support a possible role during tooth formation.
Weaknesses: While the authors have commendably performed a number of experiments to determine the role of Fubp1 gene during tooth morphogenesis, the manuscript, on the whole, appears to be a jumbled mass of data that are at times difficult to put into context and follow. Several of the data lack details that bring into question whether the results were significant or not. One gets the impression that the changes assayed were not very strongly different between control and experimental groups. However, there are merits to the study and improvements should be made in the presentation of data. This manuscript should be re-organized, with presentation of data in a logical and sequential manner based on the wealth of knowledge of tooth morphogenesis.
- Is there a way for the authors to present their results/data with clearer explanations and conclusions in each experimental set? That would help to put the data in context of the larger picture of what is known about the developmental processes during tooth morphogenesis. As currently presented, the gene and protein expression analyses (be it in situ, immunohistochemistry or RT-qPCR) jumped from one gene to the next without a clear presentation/explanation of what was of interest and what the data meant. For instance, why use ROCK1 and phalloidin? Without a developmental and cellular context and rationale based on the known molecular processes during tooth morphogenesis, the results are difficult to understand.
Overall, the findings and description of results need to be put into context. Essentially, if correctly interpreted, the authors concluded that the Fubp1 gene is involved in the differentiation of dentine and enamel – can that angle be used to drive the story to make it more focused? One could then present the data in the context of the known stages of tooth development and using the relevant markers to determine the outcome, in a sequential manner. A flow diagram of known markers of tooth development and how Fubp1 fits into the known events will make this work much more understandable.
- Description of the results in a quantitative language is oftentimes misleading and not based on actual quantitation, or if done, not presented clearly. References were made several times to quantitative changes, e.g, “decreased reactions against c-Myc” (line 88); Fig. S3 – “weaker localization of NESTIN..” (not obvious), thinner dentine layer could be due to plane of cut; Fig S4, with regard to changes in crown height, how many teeth were measured? And how was crown height assessed? Perhaps Fig. S1 should be included in Fig 2 to strengthen the quantitative assertions that there were differences in the number of c-Myc and Ki67 cells in the IEE and OEE, respectively. In Fig. 5, It is difficult to believe that the expression of Amelx and Dspp is “weaker” – perhaps reduced to a smaller area?
- The siRNA work is a knockdown, not a loss-of-function. Please edit the relevant parts.
- Discussion section is long and rambling. 1st paragraph is not necessary. Shorten and make more concise.
- Methods and materials – Clearer explanations of some parts of the methodology need to be provided: For example,
- It is not clear whether all the data presented in the manuscript were performed on organ transplants only or some from mouse embryos.
- How was the knockdown done? Was only 1 siRNA used? When control was mentioned as being used in your assay, e.g., in Fig. 5, what was the control? The scrambled RNA?
- How many replicates of all the experiments were done? For example, Fig. 1 show expression analysis in E14gene expression analyses.
- For RT-qPCR data, not clear why bars are shown for Hprt, since it is used to normalize the expression of the experimental genes.
Author Response
This study aims to analyze the developmental function of FUSE binding protein 1 (Fubp1) gene during murine tooth morphogenesis. The authors used gene expression analyses (RT-qPCR, in situ hybridization), an in vitro loss-of-function tooth organ model and renal transplantation. They found that Fubp1 was expressed in the enamel organ and condensed mesenchyme during normal tooth formation. When the Fubp1 gene was knocked down, there were changes in the expression of tooth development related gene such as Bmps and Fgf4, localization patterns of c-Myc and cell proliferation in epithelium and mesenchyme, related with altered tooth morphogenesis, decreased amelogenin and dentin sialophosphoprotein expressions and disrupted enamel rod and interrod formation in 1- and 3-week renal transplanted teeth respectively. The authors suggest that Fubp1 modulates cell physiology of dentinogenesis and amelogenesis by regulating expression pattern of signalling molecules during mice molar development to achieve the proper structural formation of hard tissue matrices and crown morphogenesis.
Strengths: The expression analyses appear to be well done, with well executed gene expression analyses. The use of an organ transplant, knockdown technology and kidney transplantation also provided additional data to support a possible role during tooth formation.
Weaknesses: While the authors have commendably performed a number of experiments to determine the role of Fubp1 gene during tooth morphogenesis, the manuscript, on the whole, appears to be a jumbled mass of data that are at times difficult to put into context and follow. Several of the data lack details that bring into question whether the results were significant or not. One gets the impression that the changes assayed were not very strongly different between control and experimental groups. However, there are merits to the study and improvements should be made in the presentation of data. This manuscript should be re-organized, with presentation of data in a logical and sequential manner based on the wealth of knowledge of tooth morphogenesis.
- Is there a way for the authors to present their results/data with clearer explanations and conclusions in each experimental set? That would help to put the data in context of the larger picture of what is known about the developmental processes during tooth morphogenesis. As currently presented, the gene and protein expression analyses (be it in situ, immunohistochemistry or RT-qPCR) jumped from one gene to the next without a clear presentation/explanation of what was of interest and what the data meant. For instance, why use ROCK1 and phalloidin? Without a developmental and cellular context and rationale based on the known molecular processes during tooth morphogenesis, the results are difficult to understand.
: As reviewer suggested, we briefly prepared the introduction of tooth development with proper references in the introduction part to avoid these misleading, especially with the lack of background information on tooth morphogenesis and development itself. In addition, we newly prepared the more detailed information on purpose of our study in the introduction part.
Page 2: Line 47-49
Following epithelial invaginations, through the combined interactions of signalling molecules and transcription factors, morphogenesis proceeds with bud, cap and bell stages at mouse embryonic stages E13, 14 and 16 respectively [6,7].
Page 2: Line 54-56
For proper modulation of these paracrine factors, it is necessary to understand the developmental roles of transcriptional regulators, which modulate the expression of target genes and contribute to ensure proper signalling regulations during organogenesis [2–4].
In addition, to announce the rationale and logical reasons why we did employ and examine those experimental tools and signaling molecules for evaluating the developmental function of Fubp1 during tooth development, we added references in each sector of results.
Page 2: Line 84-85
In order to define the cellular physiology regulated by this gene, we examined the precise localization patterns of c-Myc and Ki67, cell proliferation marker, through immunohistochemistry (Fig. 2).
Page 2: Line 95 to Page 3: Line 96-97
In addition, epithelial cell rearrangement, one of the important cellular events for epithelial morphogenesis, was examined using immunostaining against E-cadherin and ROCK1, and phalloidin staining, as were examined in previous study [19].
Page 5: Line 131-133
Tooth development is regulated with harmonious interactions among signalling molecules [6, 20], and therefore, most of the tooth development-related signalling molecules were examined using qPCR to evaluate the knock-down effects of Fubp1.
Page 5: Line 154-156
We employed renal capsule transplantation and examined the morphological alterations of the tooth to define the knock-down effect of Fubp1 at the cap stage of tooth development, as were examined in previous report [20].
- Overall, the findings and description of results need to be put into context. Essentially, if correctly interpreted, the authors concluded that the Fubp1 gene is involved in the differentiation of dentine and enamel – can that angle be used to drive the story to make it more focused? One could then present the data in the context of the known stages of tooth development and using the relevant markers to determine the outcome, in a sequential manner. A flow diagram of known markers of tooth development and how Fubp1 fits into the known events will make this work much more understandable.
: As reviewer knows well, an organogenesis is continuous and critical event which is regulated by spatiotemporal expressions of signaling molecules and the tooth development also shared the similarities in developmental processes with a range of organ development. Our study did focus on elucidating the developmental function of Fubp1, especially at cap stage. Because this cap stage of tooth development showed the most of the important signaling modulations with cellular physiology (Developmental biology; 304(2), 499-507., Histochemistry and cell biology, 144(4), 377-387) and consequently it forms hard tissue matrices and calcified morphogenesis as it develops. Also, this cap stage would provide the accumulation of abundant data in morphological and molecular aspects so far. Based on these reasons, we firstly examined the detailed developmental function of Fubp1 at cap stage to understand the transcriptional regulations of Fubp1 for cellular physiology which was not examined in the developmental biology field yet. In addition, as reviewer mentioned that, the amelogenesis and dentinogenesis are important processes for tooth development, however, these procedures are intimately related with the cap stage events such as proliferation and differentiation of epithelium and mesenchyme. Our prepared results are mostly examined the detailed function analysis at cap stage and consequent morphological changes are shown in later stage for hard tissue matrices formation. To avoid this misunderstanding, we prepared the purpose of this study in the introduction part.
Page 2: Line 47-49
Following epithelial invaginations, through the combined interactions of signalling molecules and transcription factors, morphogenesis proceeds with bud, cap and bell stages at mouse embryonic stages E13, 14 and 16 respectively [6,7].
Page 2: Line 54-56
For proper modulation of these paracrine factors, it is necessary to understand the developmental roles of transcriptional regulators, which modulate the expression of target genes and contribute to ensure proper signalling regulations during organogenesis [2–4].
Briefly, our results showed the concrete morphological changes after functional analyses in cap stage. Based on these results, we would conclude that Fubp1 in cap stage would affect in tooth development especially with amelogenesis and dentinogenesis. The spatiotemporal specific function of Fubp1 at cap stage might determine the initiation of differentiation in secretory stage because the functional study using siRNA transfection showed the altered expression and localization patterns of signaling molecules including Bmp2, Bmp4, B-catenin, Lef1, Shh, Rock1 and Fgf4. Especially, our result showed that Fubp1 upregulates the mesenchymal and outer enamel epithelium (OEE) related signaling molecules: Bmp2, Bmp4 and Rock1, whereas, downregulates the inner enamel epithelium (IEE) related signaling molecules: B-catenin, Lef1, Fgf4 and Shh. Because of these signaling alterations, there was alteration in the cellular proliferation and c-Myc localization along the IEE and OEE after Fubp1 knockdown. Therefore, it can be speculated that these altered expression and localization patterns would result the disruption in the initiation of amelogenesis and dentinogenesis for proper structural formation of tooth morphogenesis as in previous report (J Cell Physiol. 2019, 234: 20354-20365). As we only checked the functional analysis of Fubp1 through transient knockdown at cap stage, so we did RT-qPCR, immunohistochemistry (Ki67, c-Myc, ROCK1, E-cadherin), in situ hybridization (Bmp4 and Fgf4) and phalloidin staining in E14 plus 2 days tissue to get more obvious function of Fubp1 during tooth development.
In addition, as reviewer pointed out about the logical data presentation during manuscript preparation, first we checked whether Fubp1 expressed in in vivo tooth development or not (Figure 1). After observing its precise expression patterns during cap stage, we did functional analysis using siRNA knockdown system through in vitro organ cultivation (Fig. 2) and the knockdown tooth germ were subjected for immunohistochemistry and RT-qPCR analysis to check the knockdown effect of Fubp1 in the expression and localization pattern of tooth-related signaling molecules (Fig 2-5). In addition, we did transient knockdown at cap stage and checked the cellular events including proliferation and epithelial rearrangement. The logical and rationale plot, which we prepared in this study, would be enough to persuade the readers as what we want to argue.
However, although it is strongly required to reveal the direct and time-tracing evidences precise function of Fubp1 in tooth morphogenesis between cap to secretory stages, in this moment, due to the limitation of experimental techniques, it is difficult to reveal those results. In addition, any of knockout animals did not provide the function of Fubp1 in tooth development so far.
Overall, we did conclude that Fubp1, a transcription regulator, would involve in modulation of cellular events with spatiotemporal specific interaction manners. Particularly, we did examine the signaling interactions of Fubp1 with well-known and established genes first. Also, this functional evaluation of Fubp1 is the first attempt in the field of developmental biology (not in the neuroscience and immunology), so we do not prepared the flow diagram which might mislead and give biased conclusion of its function. We hope reviewer would understand our intent.
- Description of the results in a quantitative language is oftentimes misleading and not based on actual quantitation, or if done, not presented clearly. References were made several times to quantitative changes, e.g, “decreased reactions against c-Myc” (line 88).
: As reviewer pointed out, we have modified the sentences in result and discussion part to make it clear for better understanding.
Page 2: Line 87-89
Particularly, more number of c-Myc positive cells are observed in the IEE, whereas, less number are observed in the OEE of the Fubp1 knock-down specimen (Fig. 2d, f, i; Supplementary Figure S1).
Page 7: Line 210-212
Increased proliferating cells along the IEE and decreased proliferating cells along the OEE in Fubp1 knock-down tooth (Fig. 2) implied that the modulating role of Fubp1 is important along the entire enamel epithelium.
- S3 – “weaker localization of NESTIN” (not obvious), thinner dentine layer could be due to plane of cut
: As reviewer pointed out, the thinner dentin layer at one week renal calcified teeth might be due to plane of cut, however, the localization of NESTIN in the Fubp1 knock-down specimen was abrogated compared to control. The odontoblast layer and its processes showed well localized NESTIN in the control, which is abrogated in the Fubp1 knock-down specimen. Therefore, in the “Results”, “Discussion” and the “Figure legend” of supplementary figure 3, we have corrected the sentences as abrogated and weaker localization of NESTIN.
Page 6: Line 161-163
Similarly, the Fubp1 knock-down tooth revealed abrogated and weaker localization of NESTIN in secreting odontoblasts and its processes relative to the control (Supplementary Figure S3).
Page 6: Line 167-169
Although there were no significant differences in the overall dentin thickness between the control and knock-down specimen, the Fubp1 knock-down exhibited altered dentin and enamel patterned arrangements.
Back matter Page 3
Supplementary Figure S3. One-week renal calcified tooth. Masson’s trichrome staining showing one week renal calcified teeth (a, c). The localization of NESTIN is abrogated and weaker along the odontoblast layer and its processes in the Fubp1 knock-down tooth (arrow, d) when compared with the control (b). Square box indicates an enlarged view (a-d). De; dentin, Od; odontoblast. Scale bars: 50 μm (a-d).
- Fig S4, with regard to changes in crown height, how many teeth were measured? And how was crown height assessed?
: We used five different samples (number = 5) to measure the crown height of calcified teeth. The image used in the manuscript is the representative one which showed the standard features among them. We have mentioned the number of teeth which we measured for crown height in the legend of Supplementary figure 4.
Back matter Page 3
Supplementary Figure S4. Altered crown height in Fubp1 knock-down tooth. Three-week renal calcified tooth showing decreased crown height in the Fubp1 knock-down tooth compared with control (a-c). Scale bars 100 μm (a- b). Statistical analysis shows the decreased crown height in the Fubp1 knock-down tooth compared with the control (N=5) (c).
- Perhaps Fig. S1 should be included in Fig 2 to strengthen the quantitative assertions that there were differences in the number of c-Myc and Ki67 cells in the IEE and OEE, respectively.
: As reviewer suggested, we have newly prepared Figure 2 with Figure S1.
- In Fig. 5, it is difficult to believe that the expression of Amelx and Dspp is “weaker” – perhaps reduced to a smaller area?
: As reviewer pointed out, there are no obvious differences in the expressions of Amelx and Dspp in 1-week renal calcified teeth, perhaps reduced to smaller area. However, the expression patterns are comparatively weaker especially at cusp region only (indicated by arrows in Fig. 5e, f) in the Fubp1 knock-down specimen. We indicated these alterations in the result part with careful.
Page 6: Line 158-161
In addition, the Fubp1 knock-down tooth showed relatively weaker expressions of Dspp and Amelx mRNA along the cusp area when compared with the equivalent region of the control (Fig. 5b–c, e–f). The expression pattern was also reduced to smaller area in Fubp1 knock-down specimen compared to control (Fig. 5b–c, e–f).
- The siRNA work is a knockdown, not a loss-of-function. Please edit the relevant parts.
: As reviewer suggested, we have corrected this in the relevant parts through the manuscript.
Page 1: Line 21-24
In this study, to reveal the precise developmental function of Fubp1, we examined the detailed expression pattern and developmental function of Fubp1 during tooth morphogenesis by RT-qPCR, in situ hybridization and knock-down study using in vitro organ cultivation methods.
- Discussion section is long and rambling. 1st paragraph is not necessary. Shorten and make more concise.
: As reviewer suggested, we edited the first paragraph of discussion part and merged first and second paragraph to make the manuscript more concise.
Page 7: Line 191-219
Copious studies with experimental models, such as cell lines and animals models, have already been undertaken to identify the principle mechanisms of tooth development [19], however, there is still poor understanding of the key modulations among the candidate molecules in organogenesis processes. In this study, we firstly examined the precise expression pattern of a eukaryotic transcription regulator gene, Fubp1, along the epithelium and mesenchyme of the developing tooth (Fig. 1d–f). Our results showed the co-expression pattern with Hoxc12, zfp36I1, Taf10, Cnbp and Ehmt2 [20] as candidate molecules involved in directing the tooth morphogenesis via modulating the signalling molecules during embryogenesis. Particularly, in this study, as the Fubp1 knock-out mouse is embryonically lethal [21], we tried to evaluate its crucial functional roles through transient Fubp1 knock-down and in vitro organ cultivation system. Previous reports mention that c-Myc, a eukaryotic transcription regulator, is involved in cellular proliferation and differentiation [22–24]. It is also known that Fubp1 regulates the cell cycle progression through both c-Myc- dependent [25,26] and -independent [27] mechanisms. Additionally, cellular events such as proliferation and differentiation are very important during organogenesis. Our study showed that knocking down Fubp1 at E14 for 2 days increased cellular proliferation along the IEE between the secondary enamel knot-forming region (Fig. 2e, h), indicating that the IEE cells are still proliferating, whereas the control cells are already set for differentiation. The rate of cellular proliferation usually decreases as cells differentiate [28], and so it seems that Fubp1 enhances the differentiation of the IEE specifically at the cusp-forming region. The region-specific cellular proliferation after Fubp1 knock-down draws particular attention during in vitro organ cultivation. Increased proliferating cells along the IEE and decreased proliferating cells along the OEE in Fubp1 knock-down tooth (Fig. 2) implied that the modulating role of Fubp1 is important along the entire enamel epithelium. We hypothesized that these altered cell proliferation patterns would result from the altered localization pattern of c-Myc (Fig. 2d, g). However, Fubp1 knock-down tissue did not show any obvious change in the c-Myc protein expression when examined by Western blot (Supplementary Figure S3), in agreement with previous report [27]. Interestingly, immunohistochemistry analysis revealed a disrupted patterned arrangement of positive reactions against c-Myc in the IEE and OEE (Fig. 2d, g). These results suggest that there would be altered tooth morphogenesis through changed cellular events with c-Myc expression.
- Methods and materials – Clearer explanations of some parts of the methodology need to be provided: For example, it is not clear whether all the data presented in the manuscript were performed on organ transplants only or some from mouse embryos.
: As reviewer inquired, we used both in vitro cultivated specimens and in vivo tooth germs for experiment. We used in vitro tooth germ (E14+2 days cultivation) for immunohistochemistry and renal calcified teeth (All figures, except Figure 1). On the other hand, we used ICR in vivo embryos: E14, E15 and E16 for collection of tissue to check the expression pattern of Fubp1 (Fig. 1). We have also discussed this in “Materials and methods” with careful.
Page 9: Line 293-294
ICR mouse embryos were obtained from time-mated pregnant mice that were maintained in an optimal environment.
Page 9: Line 304-305
The embryonic mice molar tooth germs at E14 were cultivated for 1 and 2 days, and transplanted into the kidney capsule, as described by [37,49].
- How was the knockdown done? Was only 1 siRNA used? When control was mentioned as being used in your assay, e.g., in Fig. 5, what was the control? The scrambled RNA?
: The knockdown was done according to the siRNA transfection protocol based on Origene as described previously (Cell Tissue Res. 2014, 358, 109–121; Dev. Biol. 2007, 304, 499–507). In all figures (including Figure 5), the control means the scrambled control. The details of scrambled control were pepared in “Materials and methods”.
Page 9: Line 303-310
siRNA transfection, in vitro organ cultivation and kidney transplantation
The embryonic mice molar tooth germs at E14 were cultivated for 1 and 2 days, and transplanted into the kidney capsule, as described by [35,48]. During cultivation, embryonic tooth specimens were transfected with scrambled siRNA (negative control) (Cat# SR30004, Origene Technologies Inc., USA), Fubp1 siRNA (Cat# SR418703, Origene Technologies Inc., USA) using siTran transfection reagent (Cat# TT30001, Origene Technologies Inc., USA), in separate culture dishes for 1 day in Opti-MEM (Cat# 31985-070, Gibco, USA). The siRNAs were used at a final concentration of 100 nM. The naive and scrambled control did not show any differences in the expression of the gene (Data not shown).
- How many replicates of all the experiments were done? For example, Fig. 1 show expression analysis in E14 gene expression analyses.
: We did all the experiments at least thrice. The figures included in the manuscript are the representative one. We have mentioned this in “Materials and methods”.
Page 9: Line 297-298
All experiments described in this study were performed for 3 or more times independently.
- For RT-qPCR data, not clear why bars are shown for Hprt, since it is used to normalize the expression of the experimental genes.
: As reviewer knows well, Hprt is used to normalize the expression of the experimental genes for RT-qPCR. We used scale bars in Hprt also to show the uniformity during illustration of graph as in previous reports (J Cell Physiol. 2019, 234: 20354-20365; Cell Tissue Res. 2014, 358, 109–121).
Reviewer 3 Report
the present work is interesting and the manuscript is generally well-written. The verification method is reasonable and the revealed results are valuable for futhure research.
Author Response
Thank you for positive decision
Round 2
Reviewer 1 Report
I am satisfied with the corrections made by authors and recommend the publication of the article after any necessary proofreading
Author Response
Thank you so much for positive considerations
Reviewer 2 Report
Overall, there has not been any serious attempt to improve the manuscript based on the comments in the previous review. Several concerns still persist:
Background –
The revised version is not significantly improved. In general, the authors continue to make broad generalized statements about tooth development with no details. Examples include the following:
Line 47 , line 54 – What are the important events during bud, cap and bell stages at mouse embryonic stages E13, E14, and E16? Give some context to the biological processes you are assaying when you are using these stages of tooth development. For example, if you analyze E16 (bell stage), are you assaying for the presence of differentiation of odontoblasts, etc? That is, what are the biological processes that are happening? Put these into context to allow the understanding of the developmental events that are affected by Fubp1.
Phrases like Fubp1 modulating “cell physiology of dentinogenesis and amelogenesis” (line 32), “well-studied signalling network” (line 710), “harmonious interactions” (line 131), “harmonious cell physiology” (line 274) do not contain enough depth or details for a scientific journal. What are the interactions and the key players in the network? Do these signaling molecules fit into the bigger picture of the formation of a tooth? Are you using the gene expression analyses of key players to determine the outcomes of your experiments to point to the direction of amelogenesis and dentinogenesis? Where do the Bmps and the Fgfs come in? Where do all these molecules come into play with regard to proliferation, differentiation of enamel and dentine, patterning of the enamel and dentin? If the main conclusions are that Fubp1 is involved in dentinogenesis and amelogenesis, make that the focus of manuscript right from the outset.
The results are still not clearly rationalized in terms of the use of specific gene markers. For example, in line 85, it is still not clear why you examined c-Myc expression and how one can conclude that “altered tooth morphogenesis” is due to “changed cellular events with c-Myc expression” (line 216). What is the role of c-Myc during tooth morphogenesis? Why ROCK1? Nestin?
Figure 2 is missing h-j
Author Response
Reviewer 2
Overall, there has not been any serious attempt to improve the manuscript based on the comments in the previous review. Several concerns still persist:
: First of all, we have to point out the troublesome in the revision system. We have prepared and uploaded the new files. However, newly prepared stuffs were not evaluated properly by reviewer and editor, especially the figures of “Figure 1, Figure 2 and Figure 5”. In this revision, we did pay more careful and serious attention to improve the quality of manuscript.
Background
The revised version is not significantly improved. In general, the authors continue to make broad generalized statements about tooth development with no details. Examples include the following:
Line 47, line 54 – What are the important events during bud, cap and bell stages at mouse embryonic stages E13, E14, and E16? Give some context to the biological processes you are assaying when you are using these stages of tooth development. For example, if you analyze E16 (bell stage), are you assaying for the presence of differentiation of odontoblasts, etc? That is, what are the biological processes that are happening? Put these into context to allow the understanding of the developmental events that are affected by Fubp1.
: As reviewer suggested, we have added the information about the important biological events during mouse tooth developmental stages in the manuscript including proper references. In addition, we also added the reasons of selecting these developmental stages in our study.
Page 2: Line 47-52
Following epithelial invaginations, through the combined interactions of signalling molecules and transcription factors, tooth development proceeds with bud, cap and bell stages at mouse embryonic stages E13, 14 and 16 respectively, which shows typical morphologic features of epithelial appendages with epithelial invaginations at E13, signalling centre formation with specific epithelial structural formation of enamel knot at E14, and functional differentiation of odontoblasts and ameloblasts at E16 [6,7].
Page 2: Line 72-84
During odontogenesis, the spatio-temporal regulation of genes at cap and bell stages play crucial roles for functional and structural formation of tooth [6]. In this study, the functional evaluation of Fubp1, a transcriptional regulator gene, is performed at cap stage through siRNA knock-down and in vitro organ cultivation system to evaluate the developmental role of Fubp1 prior to differentiation of odontoblasts and ameloblasts. Because copious reports revealed that complex interactions of signalling molecules at cap stage with precise transcriptional regulations would be resulted into tooth morphogenesis especially with hard tissue matrix formation at bell stage [6-9]. In this study, as a pioneer work, we attempted to examine the detailed function of Fubp1 in tooth development at cap stage, which shows dynamic differentiation sequences with a well-studied signalling network for tooth morphogenesis. Since functional evaluation of transcriptional regulators, even though their importance in development, was difficult to perform in knockout animal models [3,6,9,23], this in vitro organ cultivation approach would be the plausible model system to provide the fine-tuning of transcription regulations by Fubp1.
Phrases like Fubp1 modulating “cell physiology of dentinogenesis and amelogenesis” (line 32), “well-studied signalling network” (line 710), “harmonious interactions” (line 131), “harmonious cell physiology” (line 274) do not contain enough depth or details for a scientific journal. What are the interactions and the key players in the network? Do these signalling molecules fit into the bigger picture of the formation of a tooth? Are you using the gene expression analyses of key players to determine the outcomes of your experiments to point to the direction of amelogenesis and dentinogenesis?
: As reviewer pointed out, we have corrected the phrases which do not contain enough depth and we have prepared the sentences in the relevant part of manuscript for better understanding. In addition, we did prepare the schematic diagram for wrapping up the developmental function of Fubp1 with other signalling molecules and cellular events in Supplementary figure S5.
Page 1: Line 32-35
Thus, our results suggested that Fubp1 plays a modulating role during dentinogenesis and amelogenesis by regulating the expression pattern of signalling molecules to achieve proper structural formation of hard tissue matrices and crown morphogenesis in mice molar development.
Page 2: Line 78-84
In this study, as a pioneer work, we attempted to examine the detailed function of Fubp1 in tooth development at cap stage, which shows dynamic differentiation sequences with a well-studied signalling network for tooth morphogenesis. Since functional evaluation of transcriptional regulators, even though their importance in development, was difficult to perform in knockout animal models [3,6,9,23], this in vitro organ cultivation approach would be the plausible model system to provide the fine-tuning of transcription regulations by Fubp1. Page 5: Line 144-146
Tooth development is regulated with the complex interactions of various signalling molecules [6, 20], and therefore, most of the tooth development-related signalling molecules were examined using qPCR to evaluate the knock-down effects of Fubp1.
Page 8: Line 299-303
Especially during enamel and dentin structural formation, the patterned array would result from balanced cellular events of dental cells, including proliferation and differentiation, in a spatio-temporal manner. However, the knock-down of Fubp1 disrupted the patterned array of those dental tissues through alterations in signalling events including proliferation and adhesion (Fig. 2, 3 and Supplementary Figure S5).
Supplementary Figure S5
Supplementary Figure S5. Schematic diagram for developmental function of Fubp1 at cap stage. Purple color indicates expression pattern of Fubp1 in developing tooth germ at E14. Arrows and blunt arrows indicate the activation and inhibition respectively.
Where do the Bmps and the Fgfs come in? Where do all these molecules come into play with regard to proliferation, differentiation of enamel and dentine, patterning of the enamel and dentin? If the main conclusions are that Fubp1 is involved in dentinogenesis and amelogenesis, make that the focus of manuscript right from the outset.
: As reviewer pointed out about the Bmp and Fgf signalling, we have discussed it in the discussion part. In addition, we did prepare the schematic diagram to illustrate the developmental function of Fubp1 with other signalling molecules (including Fgf4 and Bmps) in Supplementary Figure S5.
Page 8: Line 249-255, Page 9: Line 256-262
A copious studies reported that various paracrine factors involved in tooth morphogenesis including Bmps, Fgfs, Shh, and Wnts [3,6-9]. Among these signalling molecules, Fgfs and Shh are well known enamel knot related signalling molecules and putatively play important roles in odontoblast differentiation and tooth morphogenesis [3,6,9]. In our study, 3-week Fubp1 knock-down renal calcified teeth showed the altered crown height (Supplementary Figure S4) and based on these morphological alterations, we decided to examine expression patterns of signalling molecules which would be involved in the morphological alterations, particularly with Bmp2, Bmp4, and Fgf4 [3,6,9] (Fig. 4). The knock-down effect of Fubp1 up-regulated Fgf4 and altered the crown height (Fig. 4, Supplementary Figure. S4), however, similar expression pattern of Fgf4 resulted into long cusp height in gerbils [35] suggesting that Fgf signalling not only to cuspal height morphogenesis but also with crown height. Our study showed down-regulation of Bmp2 and Bmp4 after Fubp1 knock-down during the bell stage. This result suggests that in order to rescue weak Bmp signalling from the mesenchyme, Fgf4 would be up-regulated during reciprocal interaction between the epithelium and mesenchyme as in a previous report [38].
The results are still not clearly rationalized in terms of the use of specific gene markers. For example, in line 85, it is still not clear why you examined c-Myc expression and how one can conclude that “altered tooth morphogenesis” is due to “changed cellular events with c-Myc expression” (line 216). What is the role of c-Myc during tooth morphogenesis? Why ROCK1? Nestin?
: As reviewer suggested, we have added some information and rewrote some sentences in the results and discussion part of manuscript for better understanding the contents. As reviewer knows well, c-Myc is one of the regulator gene that code for the transcription factor and the Fubp1 is one of the potential c-Myc regulator, the knockdown of which leads to alteration in the c-Myc related gene transcription (Biochem. Biophys. Res. Commun. 2017, 491, 1047–1054; J. Exp. Clin. Cancer Res. 2018, 37, 224). The alteration of tooth morphogenesis in this study would therefore related to knock-down effect of Fubp1 (c-Myc modulator) during cap stage of tooth development. As reviewer inquired, we can speculate the synergistic role of c-Myc and Fubp1 during tooth morphogenesis as a modulator of transcription of tooth development related signalling molecules. We have discussed it in the result and discussion part.
Page 3: Line 95-97
In order to define the cellular physiology regulated by this gene, we examined the precise localization patterns of c-Myc (transcription regulator) and Ki67 (cell proliferation marker) through immunohistochemistry (Fig. 2).
Page 8: Line 233-236
Interestingly, immunohistochemistry analysis revealed a disrupted patterned arrangement of positive reactions against c-Myc in the IEE and OEE (Fig. 2d, g). These results suggest that knock-down of Fubp1, one of the potential regulator of c-Myc, [27,28] would result altered tooth morphogenesis through changed cellular events.
As reviewer inquired the rationale of using ROCK1 and NESTIN in the study, we have discussed it in the relevant section of result and discussion part.
Page 3: Line 106-109
In addition, epithelial cell rearrangement, one of the important cellular events for epithelial morphogenesis, was examined using immunostaining against E-cadherin and ROCK1, and phalloidin staining, as were examined in previous study [19].
Page 6: Line 176-180
To check whether the reduced expression pattern of Dspp and Amelx corresponds with the NESTIN localization, we did immunostaining of NESTIN, odontoblast differentiation marker, and result showed the abrogated and weaker localization of NESTIN in the secreting odontoblasts and its processes relative to the control (Supplementary Figure S3).
Page 9: Line 264-269
Meanwhile, the cellular adhesion and actin rearrangement play a crucial role during differentiation of ameloblasts [40]. In this study, the weak localization of ROCK1 and phalloidin staining along the IEE of the Fubp1 knock-down tooth implied that Fubp1 would modulate actin rearrangement during differentiation of ameloblasts before the formation of cusp morphogenesis because ROCK contributes to maintaining ameloblast polarity and enamel matrix secretion [40].
Figure 2 is missing h-j
: As reviewer pointed out, we have replaced the new Figure 2 containing Figures 2a-j.
The system did not provide our newly prepared figures in latest revision with proper. Already we have prepared the figures and uploaded as a separated file into system (you can also find out again). Particularly, in last time, we did upload “three newly prepared figures” (Figure 1, Figure 2 and Figure 5), however, it seems that reviewer and editor did not examine our newly prepared stuffs. For this revision, we all inserted into the main manuscript file for convenience of your evaluation.
